# Noise Matters: Optimizing Matching Noise for Diffusion Classifiers

**Yanghao Wang, Long Chen**[*]
The Hong Kong University of Science and Technology
ywangtg@connect.ust.hk, longchen@ust.hk

## Abstract

Although today's pretrained discriminative vision-language models (*e.g.*, CLIP) have demonstrated strong perception abilities, such as zero-shot image classification, they also suffer from the bag-of-words problem and spurious bias. To mitigate these problems, some pioneering studies leverage powerful generative models (*e.g.*, pretrained diffusion models) to realize generalizable image classification, dubbed **Diffusion Classifier** (DC). Specifically, by randomly sampling a Gaussian noise, DC utilizes the differences of denoising effects with different category conditions to classify categories. Unfortunately, an inherent and notorious weakness of existing DCs is *noise instability*: different random sampled noises lead to significant performance changes. To achieve stable classification performance, existing DCs always ensemble the results of hundreds of sampled noises, which significantly reduces the classification speed. To this end, we firstly explore the role of noise in DC, and conclude that: *there are some "good noises" that can relieve the instability*. Meanwhile, we argue that these good noises should meet **two principles**: 1) *Frequency Matching*: noise should destroy the specific frequency signals; 2) *Spatial Matching*: noise should destroy the specific spatial areas. Regarding both principles, we propose a novel Noise Optimization method to learn matching (*i.e.*, good) noise for DCs: **NoOp**. For frequency matching, NoOp first optimizes a *dataset-specific noise*: Given a dataset and a timestep $t$, optimize one randomly initialized parameterized noise. For Spatial Matching, NoOp trains a Meta-Network that adopts an image as input and outputs *image-specific noise offset*. The sum of optimized noise and noise offset will be used in DC to replace random noise. Extensive ablations on various datasets demonstrated the effectiveness of NoOp. It is worth noting that our noise optimization is orthogonal to existing optimization methods (*e.g.*, prompt tuning), our NoOP can even benefit from these methods to further boost performance. Code is available at https://github.com/HKUST-LongGroup/NoOp.

## 1  Introduction

Pretrained visual-language models (VLMs) learn the alignment of text and image from the text-image pairs. Thanks to the large-scale and in-the-wild training data, these discriminative VLMs gain some generalization capability and can even achieve zero-shot visual classification, *e.g.*, CLIP [1]. When encountering some rare or customized categories, some works like prompt optimization [2, 3] can also adapt CLIP with a few-shot training set efficiently. However, the CLIP models are criticized for an inherent bag-of-words problem [4, 5]. This problem leads to spurious bias, harming compositional inference and counterfactual reasoning. Thus, recent works [6, 7, 8, 9, 10] try to leverage the pretrained diffusion models for classification.

---

[*]Long Chen is the corresponding author.

39th Conference on Neural Information Processing Systems (NeurIPS 2025).

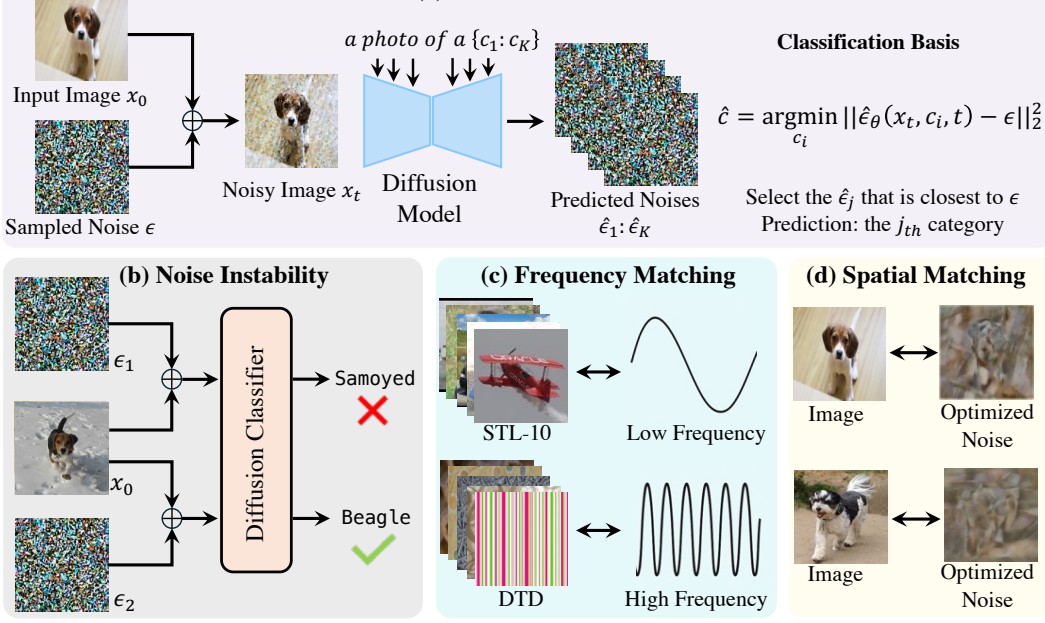

Figure 1: (a) Diffusion Classifier pipeline. (b) **Noise Instability phenomenon**: Different noises will lead to different predictions. (c) **Frequency Matching**: Good noise should destroy the specific frequency signals. (d) **Frequency Matching**: Good noise should destroy the specific spatial areas.

The typical implementation framework is **Diffusion Classifier (DC)** [6, 7]. As shown in Figure 1(a), for a given clean image $x_0$ and a specific timestep $t$, DC randomly samples a noise $\epsilon$ and follows the forward process of diffusion [11] to get the noisy image $x_t$. Then input the $x_t$ into the diffusion denoising network $\epsilon_\theta$ conditioned on $K$ different categories $\{c_i\}_{i=1}^K$ respectively and get $K$ noise predictions $\{\hat{\epsilon}_\theta(x_t, c_i, t)\}_{i=1}^K$. The category whose corresponding noise prediction has the smallest distance to the ground truth noise $\epsilon$ is the final predicted category, *i.e.*,

$$\hat{c} = \underset{c_i}{\arg\min} \|\hat{\epsilon}_\theta(x_t, c_i, t) - \epsilon\|_2^2, \quad i = 1, 2, ..., K.$$

Due to the randomness of sampled noise in DC, we can sample different noises (*e.g.*, $\epsilon_1, \epsilon_2$) for the same image $x_0$. Unfortunately, an inherent and notorious weakness of existing DC is: **Noise Instability**. As shown in Figure 1(b), different sampled noises may lead to varied performance. To mitigate this instability, all prior works [6, 7, 10] try to sample multiple noises and calculate the expectation of the distance between them and the predicted noises for classification, *i.e.*,

$$\hat{c} = \underset{c_i}{\arg\min} \, \mathbb{E}_\epsilon \left[ \|\hat{\epsilon}_\theta(x_t, c_i, t) - \epsilon\|_2^2 \right], \quad i = 1, 2, ..., K.$$

Although this ensembling strategy can somewhat relieve the instability issue, the computation overhead increases linearly with the number of noise samples, making the inference very slow (*e.g.*, one Pets image takes 18 seconds on an RTX 3090 GPU). Thus, there is a tradeoff between the unstable single-sample sampling and expensive multiple-sample ensembling for the current DC framework.

In this paper, we first try to answer a **natural question**: whether there are some *good noises* for a diffusion classifier? By "Good noise", we mean that the DC's classification results are relatively stable to different randomly sampled noises. To the extreme case, if there is a good noise, we may only need to use this noise to avoid multiple samplings, and achieve good classification results. To answer this question, we first explore and analyse the role of sampled noise in DC. Intuitively, the sampled noise destroys some parts of the image, and DC tries to find the category that can best guide the diffusion model to reconstruct the destroyed parts. Thus, the "good noise" should destroy the parts that can best reflect the difference in reconstruction effect under different categories' guidance. Therefore, we argue that good noise should meet the following **two principles**:

1) **Frequency Matching**. According to the *Frequency Bias Theory* [12, 13], given a dataset, the category-related signals are mainly in a specific frequency range. For example (*c.f.*, Figure 1(c)),

STL-10 [14] contains categories "car" and "airplane". These categories can be distinguished by their overall shapes or structures, which mainly belong to low-frequency signals. Similarly, Describable Textures (DTD) [15] has categories like "banded" and "dotted". They are mainly distinguished by high-frequency signals (*e.g.*, mutated texture). This intuitively leads to the **first principle**: *a good noise should destroy the dataset-specific frequency signals that are related to the categories.*

2) **Spatial Matching**. Given one image, the category-related signals are mainly in specific spatial areas. For example, the signals in background areas are not related to the foreground category. As shown in Figure 1(d), for a specific image, a good noise to classify it should show some similar spatial patterns to this image (*e.g.*, more noticeable damage to foreground and edges). For different images, the good noises should show different spatial layouts. This leads to the **second principle**: *a good noise should destroy the image-specific spatial areas that are related to the categories.*

In this paper, we propose the first noise optimization method by learning matching noises for DC, dubbed Noise Optimization **(NoOp)**. NoOp can effectively mitigate the *noise instability issue* by considering both frequency matching and spatial matching: 1) For frequency matching: Given a specific dataset with few-shot training samples, we randomly sample one noise as initialization. Then, directly optimize this parameterized noise based on classification loss. 2) For spatial matching: We design a Meta-Network that adopts the image (*i.e.*, $x_0$) as input and outputs an image-specific noise offset. Similarly, we optimize this Meta-Network based on classification loss. Finally, we replace the random noise of DC with the sum of the optimized noise and the noise offset.

We evaluated the effectiveness of our method over eight few-shot classification datasets. Extensive ablation results showed the stability of NoOp. Besides, we conduct empirical experiments to support the two proposed principles. Furthermore, NoOp is orthogonal to existing optimization (*e.g.*, prompt-optimization based DC), thus our NoOp can even gain extra performance gains by incorporating existing techniques. Conclusively, our contributions are summarized as follows:

- Although the noise instability of diffusion models has been widely discussed in the image generation and editing area, to the best of our knowledge, we are the first ones to study the noise instability in the discrimination task, *i.e.*, diffusion classifier.
- We propose two principles about good noise for DC: Frequency Matching and Spatial Matching, and conduct two empirical experiments to verify them.
- Regarding the Frequency Matching and Spatial Matching, we design an effective noise optimization method (NoOp). It optimizes a dataset-specific parameterized noise and a Meta-Network that can output the image-specific noise offset. By using the sum result of optimized noise and noise offset, the DC can gain stable and significant improvements across datasets.
- Extensive experiments show the effectiveness and stability of NoOp. Meanwhile, our NoOp has the orthogonal capability with other optimization methods like prompt optimization for DC. This verified that NoOp is a new few-shot learner with a unique effect. We look forward to these observations opening the door for robust generative classifiers.

## 2    Related work

**Diffusion Models (DM) for Image Classification**. As a state-of-the-art generative model, DM [11, 16] shows remarkable visual-language modeling capacity. In that case, recent studies try to unleash its discrimination potential for image classification. Studies [6, 7, 9, 8] try to leverage the vision-language knowledge of pretrained text-to-image diffusion models for zero-shot classification, while Yue *et.al.*[10] adopts the prompt optimization techniques into DM for few-shot classification. In this paper, we dive into the classification task for a new few-shot learning method of diffusion classifier.

**DM for Perception Tasks**. Beyond image classification, DM can also be used in more challenging perception tasks. From a framework perspective, Chen *et.al.* [17] leverage the DM to generate the detection bounding boxes in a refinement manner. From the pretrained knowledge perspective, studies apply the knowledge of DM in image segmentation [18, 19] and depth estimation [20].

**Noise Instability in DM.** In the image generation field, there is a widely discussed phenomenon, *i.e.*, the start point of the denoising process matters a lot to the final generation quality. To relieve this problem, recent studies can mainly be divided into three types. 1) Finding a better sampling distribution instead of the Gaussian distribution by being supervised by a third-party model [21, 22]. 2) Leveraging prior knowledge [23, 24, 25, 26, 27, 28, 29, 30] to directly refine the noise or the

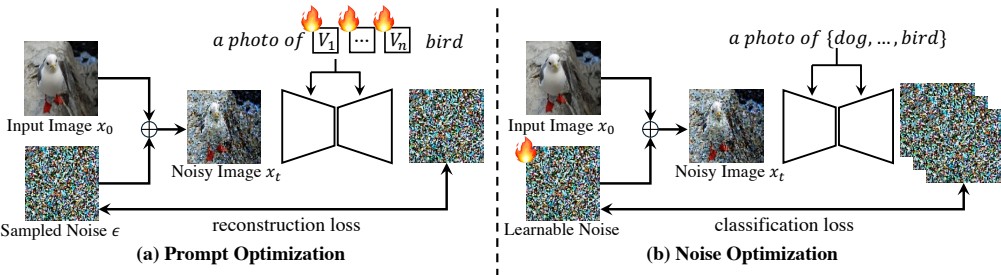

Figure 2: **Comparisons between existing prompt optimization and our noise optimization**. (a) Prompt optimization learns a few text tokens with reconstruction loss. (b) Noise optimization optimizes a parameterized noise with classification loss.

denoising path. 3) Directly optimizing the image-noise matching mechanism during the training process [31, 32]. However, we find that the noise instability also exists in the image classification task, *i.e.*, different noise leads to different classification performance. In this paper, we analyze the role of the noise and propose an optimized method to mitigate this problem.

## 3 Method

**Few-Shot Classification Formulation**. For the K-way-N-shot image classification task, typically there is a training set $\mathcal{D}$ with $K$ categories $\{c_1, ..., c_K\}$. For each category, there are $N$ labeled training samples. The few-shot learning aims at improving the model based on the training set to perform better classification on the full test set. For the diffusion model, it has multiple discrete timesteps, and each timestep controls the ratio of the original image $x_0$ and noise $\epsilon$. The timestep is input as a condition into the denoising network $\epsilon_\theta$, which means it corresponds to a $t$-specific model $\epsilon_\theta(\cdot, \cdot, t)$. Thus, for each timestep $t$, we can consider a model with a specific noise level plus a specific denoising network $\epsilon_\theta(\cdot, \cdot, t)$. In this paper, we focus on noise optimization, thus simplifying the problem by fixing the timestep at $t$ (*e.g.*, $t = 500$).

### 3.1 Preliminaries

**Diffusion Classifier (DC)**. Diffusion model (DM) [11] trains a denoising network $\epsilon_\theta$ that can reconstruct the noisy image into the original image. To be specific, given an original image $x_0$, its corresponding category text $c$, and a timestep $t$, DM first samples a noise $\epsilon$ from the Gaussian distribution, then adds it to $x_0$ to get the noisy image $x_t$ by the following forward process.

$$x_t = \sqrt{\overline{\alpha}_t}x_0 + \sqrt{1 - \overline{\alpha}_t}\epsilon, \tag{1}$$

where the $\overline{\alpha}_t$ is a $t$-related predefined parameter to control the ratio of $x_0$ and $\epsilon$. Then DM inputs $x_t$, $c$ and $t$ into the denoising network $\epsilon_\theta$ to predict the added noise, *i.e.*, $\hat{\epsilon}_\theta(x_t, c, t)$. The training objective is minimizing the MSE loss between $\hat{\epsilon}_\theta(x_t, c, t)$ and ground truth $\epsilon$:

$$\min_\theta \mathbb{E}_{\epsilon, x, c, t} \left[ ||\epsilon - \hat{\epsilon}_\theta(x_t, c, t)||_2^2 \right]. \tag{2}$$

Based on this training objective, the diffusion classifier (DC) can leverage the effect difference of different $c$ to perform image classification. Specificly, given an test image $x_0$ and $t$, DC first adds noise to get the noisy image $x_t$ (*c.f.*, Eq. (1)). Then use $K$ categories $\{c_1, ..., c_K\}$ as guidance to denoise $x_t$ respectively and get $K$ noise prediction $\{\hat{\epsilon}_\theta(x_t, c_i, t)\}_{i=1}^K$. Thus, based on Eq. (2), DC selects the category that can best denoise $x_t$ as the final category prediction:

$$\hat{c} = \underset{c_i}{\arg\min} \, \mathbb{E}_\epsilon ||\hat{\epsilon}_\theta(x_t, c_i, t) - \epsilon||_2^2, \quad i = 1, 2, ..., K. \tag{3}$$

**Prompt-Optimization of DC**. The pertaining dataset of diffusion models may have a distribution gap with the downstream classification datasets. Thus, it is hard to use a few natural words to accurately describe a category directly. To fill this gap, existing few-shot DCs [10] are mainly based on prompt optimization [2, 3]. As shown in Figure 2(a), for each category, they set $n$ learnable text tokens $\{[V_1] [V_2] ... [V_n]\}$ and add them into the text prompt to supplement the details of the category. Then they fix the whole diffusion model and only tune tokens on the few-shot training set based on Eq. (2).

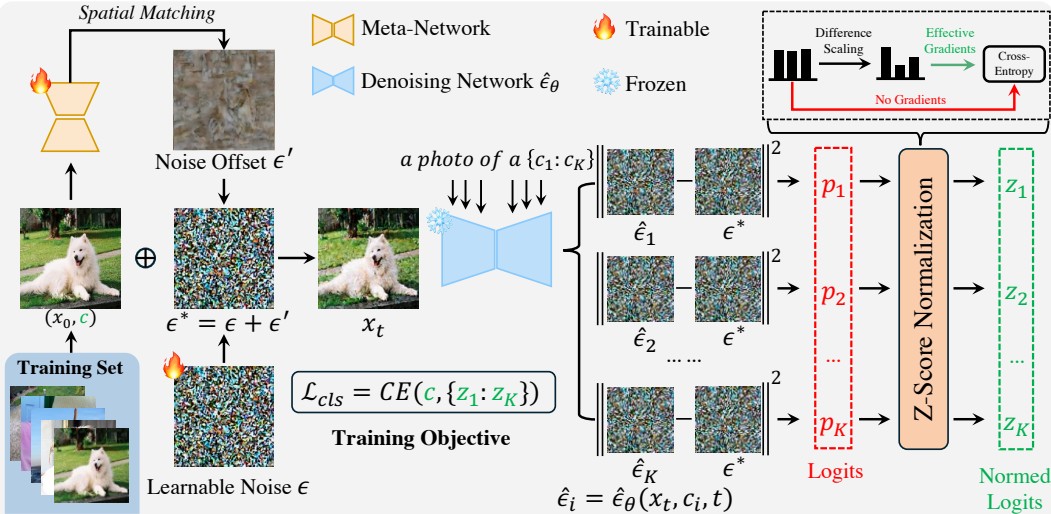

Figure 3: **Pipeline of NoOp**. Given a diffusion classifier and a few training samples, we optimize a parameterized noise ($\epsilon$) and a Meta-Network based on the classification loss. The Meta-Network is used to predict an image-specific noise offset ($\epsilon'$). Besides, we use Z-score normalization for more stable training. In inference, we use the sum of optimized noise and the offset as the final noise ($\epsilon^*$).

## 3.2 Noise Optimization

From a new perspective, our method focuses on finding a better noise that can be used in DC. As shown in Figure 2(b), we try to optimize the noise in DC for better performance on image classification. To find the guideline of good noise, we first consider the physical role of noise in DC: noise is used for destroying some signals of the original image $x_0$. If the lost signals are category-related, the denoising effect differences can emerge under different category guidance. Thus, a good noise can target the category-related signals. Specifically, the noise should meet two principles:

1) Frequency Matching: Since the sample features within one dataset usually satisfy a specific distribution, the category-related signals of the samples in the dataset will fall into a specific frequency range. For different datasets, the category-related signals are distributed in different frequencies. Thus, the noise should destroy the specific frequency signals of the target dataset. Motivated by this, we randomly initialize a noise $\epsilon$ for the whole dataset and make it learnable. This noise is expected to be optimized for destroying the category-related frequency signals.

2) Spatial Matching: Different images within one dataset have different spatial layouts. The category relevance of different spatial parts varies a lot, *i.e.*, some spatial parts of the image are category-related and some are not. Thus, the noise should have different degrees of destruction regarding different parts of the image. Motivated by this, we set a Meta-Network $U_\theta$ that adopts the original image as input and outputs an image-specific noise offset $\epsilon'$. This offset indicates the spatial adjustment with respect to $\epsilon$, which can make the noise better destroy the category-related spatial parts.

**Training.** As shown in Figure 3, firstly, we have a training set $\mathcal{D}$, a pretrained denoising network $\epsilon_\theta$, the learnable $\epsilon$ and Meta-Network $U_\theta$. The parameterized $\epsilon$ is initialized by randomly sampling from a Gaussian distribution. The Meta-Network is a light U-Net architecture containing multiple convolutional layers of up-sampling and down-sampling. For each $(x_0, c) \in \mathcal{D}$, we first input the $x_0$ into the $U_\theta$ to get the noise offset $\epsilon' = U_\theta(x_0)$. Then, we get a refined noise $\epsilon^* = \epsilon + \epsilon'$. After that, we conduct Eq. (1) to get the noisy image $x_t$ (the noise used for the forward process is $\epsilon^*$). Then we input $x_t$, $t$ and all categories $\{c_1, ..., c_K\}$ into denoising network to get $K$ noise predictions $\{\hat{\epsilon}_\theta(x_t, c_i, t)\}_{i=1}^{K}$. Based on the distance between noise predictions and ground truth noise $\epsilon^*$, we can calculate the classification logit for each category:

$$p_i = -||\epsilon^* - \hat{\epsilon}_\theta(x_t, c_i, t)||_2^2, \tag{4}$$

where the minus operator "$-$" is to convert the distance into the classification logit.

After getting the classification logits, we found that all logit values are very close. This is because they are calculated based on the distance between predicted noises and the ground truth noise. And all the distances are very close due to the diffusion model's property. Thus, if we directly optimize the $\epsilon$ and $U_\theta$ based on these logits, the optimization gradient is close to zero, making training difficult. To mitigate this problem, we use Z-score normalization [33] on the category dimension to effectively amplify the signal difference between the logit. Specifically, we first calculate the mean $\mu$ and biased variance $\sigma$ of $\{p_1, p_2, ..., p_K\}$ respectively:

$$\mu = \frac{1}{K} \sum_{j=1}^{K} p_j, \qquad \sigma = \sqrt{\frac{1}{K} \sum_{j=1}^{K} \left(p_j - \mu\right)^2}. \tag{5}$$

Then, based on the mean and variance, calculate each normalized logit:

$$z_i = \frac{p_i - \mu}{\sigma}, \quad i = 1, 2, \ldots, K. \tag{6}$$

After getting the normalized logits $\{z_1, z_2, ..., z_K\}$, we can optimize our $\epsilon$ and $U_\theta$ by minimizing the cross-entropy loss on the training set:

$$\epsilon, U_\theta = \underset{\epsilon, U_\theta}{\operatorname{argmin}} \frac{1}{KN} \sum_{(x_0, c) \in \mathcal{D}} \left[ -\log \frac{\exp(z)}{\sum_{i=1}^{K} \exp(z_i)} \right], \tag{7}$$

where $z$ is the normalized logit of the ground truth category. After training, we can get an optimized dataset-specific $\epsilon$ and a Meta-Network that can produce an image-specific noise offset $\epsilon'$. We use $\epsilon^* = \epsilon + \epsilon'$ as the final optimized noise that can meet frequency matching and spatial matching well. We use $\epsilon^*$ to replace the randomly sampled noise in DC methods for inference.

## 4 Experiments

### 4.1 Few-Shot Learning

**Settings.** To evaluate how NoOp can benefit the DC, we conducted few-shot learning experiments. Specifically, we followed the few-shot evaluation protocol of CLIP [1], using 1, 2, 4, 8, and 16 shots for training, respectively, and deploying models in the full test sets. We evaluated three diffusion models, *i.e.*, *Stable Diffusion-v1.4*, *Stable Diffusion-v1.5* [34], *Stable Diffusion-v2.0* [35] across eight datasets: *CIFAR-10* [36], *CIFAR-100* [36], *Flowers102* [37], *DTD* [15], *OxfordPets* [38], *EuroSAT* [39], *STL-10* [14] and *FGVCAircraft* [40]. We compared our NoOp with the ensembling methods [7, 6] (*i.e.*, sampling 5 noises for each image and consequently taking 5 times the computation for inference). For fair comparisons, we fixed the timestep $t = 500$. We used the Adam optimizer [41] with a $1e^{-2}$ and $1e^{-3}$ learning rates for the learnable noise the Meta-Network respectively. After training 20 epochs, we reported the top-1 accuracies. Results are averaged on three random seeds.

**Results.** From the results in Figure 4, we have two observations: 1) Though our NoOp only refines the noise (a very small-scaled part of the diffusion model), the performance improves with a relatively large scale. This demonstrated how severe the noise instability problem of DC is. 2) Across different datasets and DC versions, our NoOp can gain consistent improvements and outperform the 5-times expensive ensembling methods (*e.g.*, according to the average over eight datasets, with only two shots of NoOp can beat ensembling). This indicates that NoOp is an efficient few-shot learner.

### 4.2 Compare with Prompt-Optimization DC

**Settings.** To explore the difference between noise optimization and prompt optimization. We compared the performance of a prompt-optimization based DC method [1], *i.e.*, *TiF Learner* [10], our NoOp, and the combination of both on *FGVCAircraft* [40] and *ISIC-2019* [42]. For fairness, we used *Stable Diffusion-v2.0* [35] as the classifier. All other settings are the same as Sec. 4.1.

**Results.** Shown in Table 1, the second and third rows are the few-shot performance of *TiF Learner* and NoOp, respectively. We can see that both of them can improve the classification independently. And generally, the performance can be further improved by combining them. This indicates that NoOp is a new few-shot learner, which has a different effect from the current prompt optimization.

---

[1]The prompt-based optimization DC is introduced in Sec. 3.1

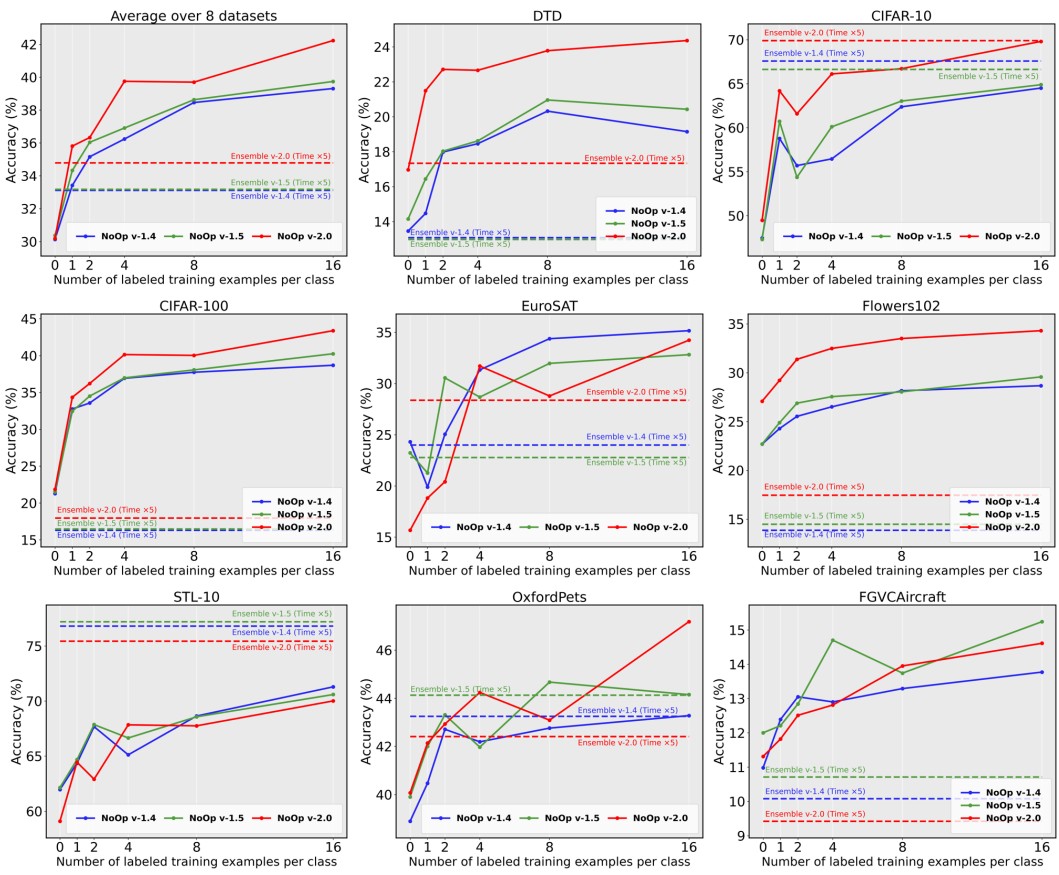

Figure 4: **Main results of few-shot learning on the eight datasets**. Overall, NoOp effectively turns DC into a strong few-shot learner, achieving significant improvements over zero-shot DC (shot=0) and generally outperforms (with only 2 shots) the expensive ensembling methods.

Table 1: Comparison of Tif Learner, NoOp, and the combination of them. Overall, both prompt optimization (TiF) and noise optimization are effective few-shot learners. Moreover, their effects are complementary, which means that they can further improve performance when used in combination.

| Method | ISIC-2019 | | | | | FGVCAircraft | | | | |
|---|---|---|---|---|---|---|---|---|---|---|
| | 1 | 2 | 4 | 8 | 16 | 1 | 2 | 4 | 8 | 16 |
| Zero-shot DC | 13.25 | 13.25 | 13.25 | 13.25 | 13.25 | 11.31 | 11.31 | 11.31 | 11.31 | 11.31 |
| TiF Learner | 17.25 | 13.89 | 19.76 | 19.91 | 17.53 | 15.60 | 17.04 | 16.98 | 19.47 | 21.03 |
| NoOp | **18.41** | **20.80** | **23.72** | 18.41 | 20.82 | 11.82 | 12.51 | 12.81 | 13.95 | 14.61 |
| NoOp + TiF | 18.32 | 19.23 | 14.45 | **29.29** | **21.59** | **15.99** | **18.54** | **19.59** | **22.44** | **25.74** |

## 4.3 Cross-Dataset Transfer

**Settings**. To demonstrate NoOp's generalization ability on open-set recognition, we evaluated our method in the cross-dataset transfer experiments. We optimized the noise and the Meta-Network on 4-shot ImageNet (source dataset) and tested it on the source dataset and eight target datasets. We compared our method with the ensembling methods (5 times computation), and $\Delta$ denotes our method's gain over ensembling.

**Results**. As shown in Table 2, the noise and Meta-Network learned on ImageNet can also be used in most of the target datasets. Specifically, six of eight can outperform the ensembling methods, and our method can improve 2.86% accuracy on average. This demonstrated that the noise optimization can learn some generalized knowledge that is beneficial to the classification, i.e., how to destroy the target part of the image and better utilize the reconstruction capacity to achieve classification. To this end, this result indicates that our method can be further leveraged in open-world scenarios.

Table 2: **Comparison of the Ensemble method in the cross-dataset transfer setting.** Noises are optimized on the source datasets (4-shot ImageNet) and applied to the eight target datasets. NoOp demonstrates good transferability across datasets. Δ denotes NoOp's gain over Ensemble

| | Source | Target | | | | | | | | |
|---|---|---|---|---|---|---|---|---|---|---|
| | ImageNet | DTD | CIFAR-10 | CIFAR-100 | EuroSAT | Flowers102 | STL-10 | OxforfPets | FGVCAircraft | *Average* |
| Ensemble (Time ×5) | 25.94 | 17.34 | **69.91** | 17.96 | 28.37 | 17.47 | **75.44** | 42.41 | 9.42 | 34.79 |
| NoOp | **26.34** | **21.70** | 63.26 | **29.36** | **29.31** | **29.48** | 71.66 | **45.90** | **10.56** | **37.65** |
| Δ | +0.40 | +4.36 | -6.65 | +11.40 | +0.94 | +12.01 | -3.78 | +3.49 | +1.14 | +2.86 |

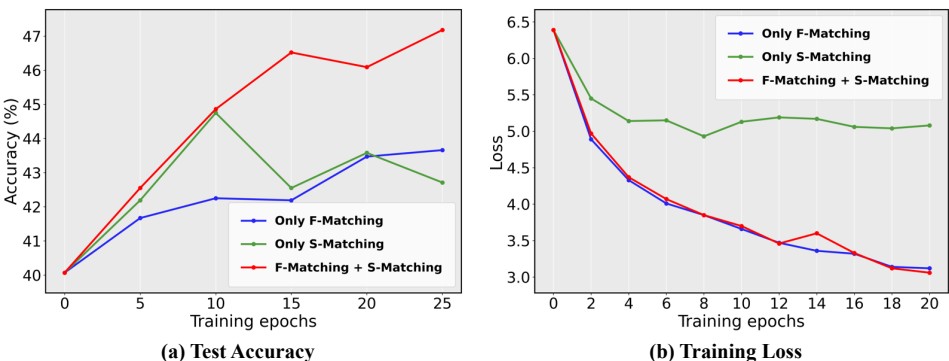

(a) Test Accuracy      (b) Training Loss

Figure 5: **Ablation Study.** The test accuracy and training loss curve across epochs of only frequency matching ("F-Matching" in blue line, *i.e.*, only optimize a dataset-specific noise), only spatial matching ("S-Matching" in green line, *i.e.*, only optimize a Meta-Network for image-specific noise offset) and combining of them (red line).

## 4.4 Ablation Study

**Settings.** To evaluate the effect of parameterized noise optimization (for frequency matching) and Meta-Network optimization (for spatial matching). We ablated the effectiveness of each component on 16-shot *OxfordPets* [38]. We used *Stable Diffusion-v2.0* [35] as the classifier and reported the top-1 accuracy and training loss across training epochs. All other settings are the same as Sec. 4.1.

**Results.** As shown in Figure 5, we have two observations: 1) Both frequency matching (F-Matching) optimization and spatial matching (S-Matching) optimization can improve the classification during training. This indicates that the two components are designed appropriately. 2) Further accuracy improvement and faster convergence are gained by combining them, demonstrating that F-Matching and S-Matching optimize the noise in two different perspectives.

## 4.5 Extend to Flow-based Diffusion Models

**Settings.** To evaluate the generalization and robustness of NoOp, we transferred it to flow-based diffusion models [43, 32]. We demonstrated the few-shot results of *Stable Diffusion-v1.5* [34] (SD-v1.5) and the *Rectified Flow* [32] (ReFlow, fine-tuned from SD-v1.5) on CIFAR-10 [36] and OxfordPets [38]. All other settings are the same as Sec. 4.1.

**Results.** As shown in Table 3, we have two observations: 1) The performance of ReFlow is better than SD-v1.5, although they have the same training set and architecture. This suggests that the flow-based diffusion model is also superior in the discriminative task, not just limited to the generative domain. 2) Our NoOp can not only improve the DDPM-based [11] SD-v1.5 but also the flow-based [32, 43] ReFlow. This indicates that our analysis of the noise role and optimization of noise is a general framework for the diffusion family.

Table 3: The few-shot results of NoOp with SD-v1.5 and the ReFlow. Our NoOp is compatible with general diffusion models.

Table 4: The computational time comparisons between ensemble and NoOp. The unit of time cost is one hour with 32 V100 GPUs.

| Shot Number | CIFAR-10 | | OxfordPets | |
|---|---|---|---|---|
| | SD v-1.5 | ReFlow | SD v-1.5 | ReFlow |
| shot = 0 | 47.30 | 70.15 | 39.90 | 56.23 |
| shot = 1 | 60.72 | 73.64 | 42.00 | 58.79 |
| shot = 2 | 54.37 | 72.92 | 43.31 | 60.63 |
| shot = 4 | 60.11 | 74.51 | 41.97 | 60.60 |
| shot = 8 | 63.03 | 76.34 | 44.67 | 62.74 |
| shot = 16 | 64.89 | 79.73 | 44.15 | 63.89 |

| Setting | Training | Inference | Total | Acc (%) |
|---|---|---|---|---|
| Ensemble | 0 | 153.0 | 153.0 | 25.94 |
| Shot = 0 | 0 | 30.6 | 30.6 | 17.34 |
| Shot = 1 | 0.4 | 30.6 | 31.0 | 24.62 |
| Shot = 2 | 0.8 | 30.6 | 31.4 | 25.31 |
| Shot = 4 | 1.6 | 30.6 | 32.2 | 26.34 |
| Shot = 8 | 3.2 | 30.6 | 33.8 | 27.59 |
| Shot = 16 | 6.4 | 30.6 | 37.0 | 29.13 |

## 4.6 Computational Overhead Analysis

**Settings**. As the original Diffusion Classifier suffers from high computational cost and slow inference speed, to verify whether our method can improve the efficiency, We used ImageNet as an example to demonstrate how our method can improve the efficiency of the original Diffusion Classifier. The unit of time cost is one hour with 32 NVIDIA V100 GPUs. For the ensemble method, we randomly sample five different noises.

**Results**. As shown in Table 4, our training time is quite smaller than the inference time. Even for the 16-shot training, the total time is only around 24% while the accuracy gains 3.19% compared with the 5-times ensembling method. This demonstrates that our method can significantly improve the overall efficiency. Additional formal cost analysis is in the Appendix E.

## 4.7 Role of Noise Validation

**Settings**. In this paper, we argue that the good noise in DC is trying to destroy the category-related signals. To validate this assumption, we compared two sets of noisy images. 1) Noisy images by adding random noise. 2) Noisy images by adding optimized noise (from NoOp). For fairness, we fixed $t = 500$ to get the noisy images. We tested on 4 datasets: *OxfordPets* [38], *DTD* [15], *EuroSAT* [39] and *Flowers102* [37], and then reported the averaged *CLIP Score* [44]. The *CLIP Score* can reflect how much category-related signals are destroyed. A lower *CLIP Score* indicates that the noise destroys more category-related signals. we used *Stable Diffusion-v2.0* [34] as the classifier and *ViT-B/32 CLIP* for *CLIP Score* calculation. All other settings are the same as Sec. 4.1.

**Results**. As shown in Figure 6(a), the *CLIP Scores* are decreased on all datasets. This indicates that our optimized noise can better destroy the category-related signals of images, which can support our analysis of the role of noise in DC.

## 4.8 Frequency Matching Validation

**Settings**. To validate our proposed frequency matching principle, we selected two representative datasets for noise optimization: 1) *CIFAR-10* [36], a coarse-grained dataset where the categories are mainly distinguished by low-frequency signals (object shape and structure). 2) *Describable Textures* (*DTD*) [15], a texture dataset where the categories are mainly distinguished by high-frequency signals (mutated texture). We randomly sampled one noise and then directly optimized it (w/o optimizing the Meta-Network) on these two different datasets, respectively. During the training process, we recorded the frequency change of this noise by calculating the *high-frequency signal ratio* of the noise. The method to get the *high-frequency signal ratio* is based on the 2D Fourier transform (*c.f.*, Appendix B). A higher *high-frequency signal ratio* means this noise is a relatively high-frequency noise.

**Results**. As shown in Figure 6(b), when training on the low-frequency dataset, *i.e.*, *CIFAR-10*, the frequency of the noise will decrease. Otherwise, when training on the high-frequency dataset, *i.e.*, *DTD*, the frequency of the noise will increase. This indicates that, the frequency of noise will move towards matching the frequency of the dataset, which verifies our analysis of noise role and the frequency matching principle.

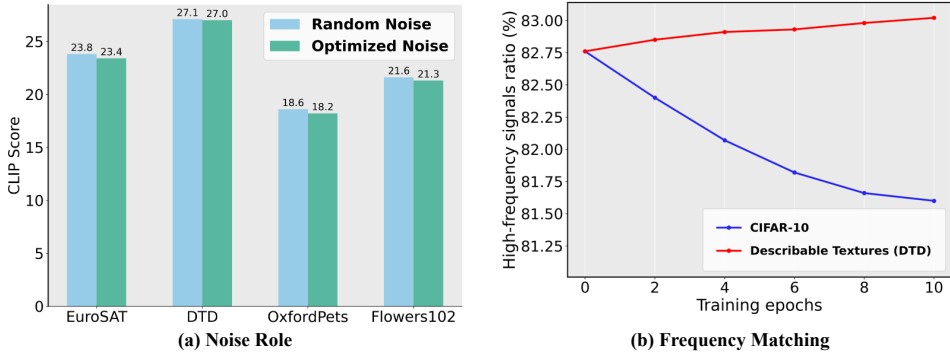

Figure 6: **Empirical Validations.** (a) The average CLIP Scores comparison of the noisy images to validate the role of good noise. (b) The high-frequency signals ratio comparison of DTD and CIFAR-10 to validate the frequency matching principle.

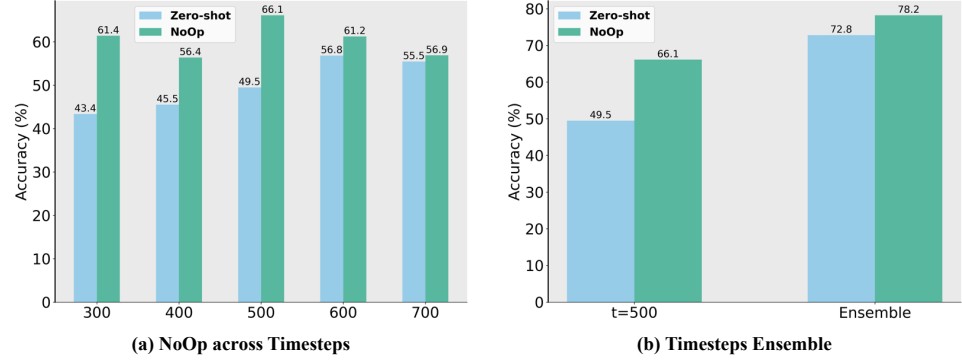

Figure 7: **Ablation study and ensemble results**. (a) The performance of NoOp across different timesteps. (b) The performance of the timesteps ensemble (5 different timesteps) of NoOp.

### 4.9 Time steps Ablation and Ensemble

**Settings**. In the main paper, we fixed the timestep at 500 to focus on the noise optimization. To evaluate whether our NoOp is consistently effective on other timesteps. We conducted a timestep ablation study. We tested $t = 300, 400, 500, 600, 700$ with *Stable Diffusion-v2.0* [35] as the pretrained diffusion model on 4-shot *CIFAR-10* [36]. Moreover, we also test the ensemble with 5 different timesteps (*i.e.*, $t = 300, 400, 500, 600, 700$) after using NoOp. The pretrained diffusion model is *Stable Diffusion-v2.0* [35] and dataset is 4-shot *CIFAR-10* [36]. We also showed the results of zero-shot w/o ensemble, zero-shot w/ensemble, and NoOp w/o ensemble for comparison.

**Results**. In Figure 7(a), our NoOp can gain consistent improvements across different timesteps. This indicates that NoOp is robust and effective across timesteps. Besides, as shown in Figure 7(b), we can see that compared with the one-timestep (*i.e.*, $t = 500$) NoOp, the timesteps ensemble can improve around 12% top-1 accuracy and outperforms the zero-shot ensemble 5.4%. This demonstrates that the ensemble strategy can also be applied for our NoOp if there are enough computational resources.

## 5 Conclusion

In this paper, we revealed the noise instability in the diffusion classifier and analyzed the role of noise. Then we proposed two principles as guidelines for designing the noise optimizing framework, *i.e.*, NoOp, to refine the random noise into a stable matching noise. Extensive experiments show that our NoOp is an effective few-shot learner. It can not only learn generalized knowledge for cross-dataset recognition and be compatible with the flow matching models. Moreover, we conducted two empirical experiments to support our proposed principles. As a highlight, we find that our NoOp is orthogonal to existing optimization methods like prompt tuning. This shows that noise optimization has a unique physical meaning and effect. In the future, we are going to extend the noise optimization into flow matching models and unified models (*e.g.*, auto-regressive model plus diffusion model).

# 6 Acknowledgements

This work was supported by the National Natural Science Foundation of China Young Scholar Fund (62402408) and the Hong Kong SAR RGC Early Career Scheme (26208924).

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

# Appendix

This appendix is organized as follows:

- Section A introduces the backgrounds of diffusion classifiers (DC), including the basic theory, implementations, and advantages compared with Vision-language Models (VLM).

- Section B gives the theory of the 2-D Fourier transform, *i.e.*, the frequency analysis method for images in our frequency-matching related experiments.

- Section C gives the justification of the NoOp's mechanism.

- Section D provides more experimental results. Firstly, we give a few additional few-shot learning results compared with the prompt-optimization DC in Section D.1. Secondly, we provide the ablation results of Meta-Network with different architectures in Section D.2. Finally, we give the qualitative results of our NoOp's stability in Section D.3.

- Section E provides the implementation details of our NoOp and the reproduction details. Besides, we also give the computation cost.

- Section F analyzes the limitations of noise optimization DC and its societal impacts.

## A    Background of Diffusion Classifiers (DC)

The recent surge of visual generation benefits from Diffusion models [11, 16], and high-quality images and videos are generated by sampling from Gaussian noise. Meanwhile, the downstream tasks include editing [45, 46], composing [47], and erasing [48] are also frequently researched.

**The Theory of DC**. DC leverages the vision-language alignment knowledge of pre-trained diffusion models, which are trained to progressively denoise noisy images through an iterative forward and reverse Markov process. Specifically, the diffusion process is defined by the forward Markov chain, progressively adding Gaussian noise to a clean image $x_0$ to produce the corresponding noisy image $x_t$:

$$q(x_t|x_{t-1}) = \mathcal{N}(x_t; \sqrt{\overline{\alpha}_t}x_{t-1}, (1 - \overline{\alpha}_t)I), \tag{8}$$

where $\overline{\alpha}_t$ determines the amount of noise added at each step. Conversely, the reverse process attempts to reconstruct the original image from the noisy version using learned denoising functions $\mu_\theta$ and $\Sigma_\theta$:

$$p_\theta(x_{t-1}|x_t) = \mathcal{N}\left(x_{t-1}; \mu_\theta(x_t, t), \Sigma_\theta(x_t, t)\right). \tag{9}$$

A diffusion classifier uses the denoising capability of the diffusion model to approximate the classification probabilities of each category. Specifically, given a clean image $x_0$, class labels $c_i$, the classifier selects the class minimizing the weighted reconstruction error as follows:

$$\hat{c} = \underset{c_i}{\operatorname{argmin}}\, \mathbb{E}_{\epsilon,t}\left[w_t\|x - \tilde{x}_\theta(x_t, c_i, t)\|_2^2\right], \tag{10}$$

where $\epsilon \sim \mathcal{N}(0, I)$, $t \sim \operatorname{Uniform}([0, T])$, and $w_t$ is a time-dependent weighting function.

To estimate the expectation efficiently, Monte Carlo sampling is employed:

$$\mathbb{E}[f(x)] \approx \frac{1}{N}\sum_{i=1}^{N} f(x^{(i)}), \quad x^{(i)} \sim q(x_t|x_0), \tag{11}$$

where each noisy sample $x^{(i)}$ is generated independently.

**The Implementations of DC**. In practice, DC implementations often utilize pretrained text-to-image diffusion models such as Stable Diffusion [34, 35]. The timestep weights can be even or varied, such as based on the signal-to-noise ratio (SNR). Another typical implementation involves computing a re-weighted reconstruction error across diffusion time-steps, selecting the class whose conditioning prompt results in minimal reconstruction error at an early time-step [10]. More efficient implementations reduce computational overhead by applying variance reduction strategies, including shared-noise sampling and candidate class pruning. Alternatively, few-shot DC learn class-specific concepts on top of pretrained diffusion models to enhance the classification of nuanced attributes, improving robustness against spurious visual correlations [10].

**The advantages of DC**. Compared with vision-language model (VLM) classifiers such as CLIP, DC exhibits several significant advantages. Firstly, DC demonstrates superior multimodal compositional reasoning and attribute binding capabilities, outperforming VLM classifiers on tasks requiring nuanced understanding of visual features and their textual descriptions [6, 7]. Secondly, DC inherently possesses robustness to texture-shape biases and spurious correlations, a common issue in discriminative few-shot training with VLMs, by utilizing a principled generative modeling approach that isolates semantic attributes effectively through diffusion time-steps [10]. Lastly, the generative modeling nature of DC allows seamless adaptation to few-shot and zero-shot learning settings without extensive retraining, making them versatile and efficient for various downstream tasks [6, 10].

## B  Theory of 2-D Fourier Transform

The two-dimensional (2-D) Fourier transform is a fundamental tool in frequency domain analysis of images. Given a spatial domain image $f(x, y)$, the continuous 2-D Fourier transform $F(u, v)$ is mathematically defined as:

$$F(u,v) = \int_{-\infty}^{\infty} \int_{-\infty}^{\infty} f(x,y) e^{-j2\pi(ux+vy)} dx dy, \tag{12}$$

where $j$ is the imaginary unit, $u$ and $v$ represent the spatial frequencies in the horizontal and vertical directions, respectively. Conversely, the inverse 2-D Fourier transform recovers the original spatial image from its frequency representation:

$$f(x,y) = \int_{-\infty}^{\infty} \int_{-\infty}^{\infty} F(u,v) e^{j2\pi(ux+vy)} du dv. \tag{13}$$

In digital image processing, discrete counterparts of these transforms are commonly utilized due to practical constraints. Given a discrete image $f(x, y)$ of size $M \times N$, the Discrete Fourier Transform (DFT) is defined as:

$$F(u,v) = \sum_{x=0}^{M-1} \sum_{y=0}^{N-1} f(x,y) \, e^{-j2\pi\left(\frac{u\,x}{M} + \frac{v\,y}{N}\right)}, \tag{14}$$

and the inverse Discrete Fourier Transform (IDFT) is:

$$f(x,y) = \frac{1}{M\,N} \sum_{u=0}^{M-1} \sum_{v=0}^{N-1} F(u,v) \, e^{j2\pi\left(\frac{u\,x}{M} + \frac{v\,y}{N}\right)}. \tag{15}$$

The Fourier transform decomposes an image into its constituent frequency components, effectively distinguishing low-frequency (smooth or slowly varying) and high-frequency (sharp edges and fine details) content [49, 50]. In our frequency matching validation (*c.f.*, Sec. 4.1), leveraging this decomposition provides valuable insights into how diffusion processes interact with different frequency components, informing analysis and manipulation strategies at various frequency scales [51]. After the above frequency decomposing technique, for each image or noise latent, we can represent its frequency by its high-frequency signals ratio. Specifically, we set a cutoff threshold (*e.g.*, 0.3 was used for Figure 7(b)) to divide all the signals into two parts: high-frequency and low-frequency. The ratio of high-frequency signals to low-frequency signals is used to represent its frequency.

## C  Justification of the mechanism of noise optimization

To justify the mechanism of noise optimization and why it works for the Diffusion Classifier, we first recall the mechanism of the Diffusion Classifier (DC).

Given an inference image, the DC firstly adds some noise to destroy some of the signals that are related to the category. Then, using different categories as the prompts to reconstruct the noisy image. The final prediction is the category that can reconstruct it the best.

In this pipeline, a very crucial part is adding noise because it decides which signals are removed by adding noise. The best situation is trying to remove the category-related signals and maintain other

category-independent signals. In that case, the reconstruction difference of diffusion models under different prompts can be fully unleashed.

Thus, our method can learn an optimized noise to target the category-related signals. This kind of optimized noise is usually in the low-probability areas of the standard Gaussian distribution, and thus is very difficult to sample. In comparison, the randomly sampled noises are usually in the high-probability areas of the standard Gaussian distribution but lack the target-destroying capacity.

We compared the random noise and the optimized noise and reported their mean, variance, and standard Gaussian distribution probability density (pdf). For the randomly sampled noise, the mean, variance, and pdf (log) are $-0.0032$, $1.0021$, and $-23265$, respectively. In contrast, for the optimized noise, they are $-0.0043$, $1.0282$, and $-23479$.

We can see that, compared with the random sampled noise, the optimized noise's mean is farther away from $1$, variance is farther away from $0$, and the probability density is lower. This demonstrated that the optimized noise is usually in the low-probability areas of the standard Gaussian distribution to target the category-related signals. And this kind of noise is difficult to sample. Thus, we obtain it by optimization.

# D  Additional Results

## D.1  Additional Few-shot Learning Comparisons

**Settings**. To further evaluate the effectiveness of our NoOp, we conducted additional few-shot (shot=1,2,4,8) learning experiments. Specifically, we compared our NoOp with two baselines, *i.e.*, ensemble (sample five different noises for one image) zero-shot DC [7, 6] and prompt learning DC [10, 2] (*c.f.*, Section 3.1) across eights datasets: *CIFAR-10* [36], *CIFAR-100* [36], *Flowers102* [37], *DTD* [15], *OxfordPets* [38], *EuroSAT* [39], *STL-10* [14] and *FGVCAircraft* [40]. We used *Stable Diffusion-v2.0* [35] as the pretrained diffusion model. Results are averaged on three random seeds.

**Results**. As shown in Figure 8, we can have two observations: 1) As two few-shot learners, both prompt learning and noise optimization can outperform expensive ensemble methods with less than 4 shots. 2) Prompt learning improves significantly on *EuroSAT*, *Flowers102* and *FGVCAircraft* while NoOp performs better on other five datasets. However, we can see that NoOp is generally a more stable few-shot learner. Because for some datasets and shot numbers, the prompt learning may decrease the performance (*e.g.*, when the shot number is 1, the performance of prompt learning is lower than zero-shot on six datasets). In contrast, our NoOp shows more stable and consistent improvements across datasets and shot numbers.

## D.2  Meta-Network Ablation

**Settings.** To ablate the Meta-Netwrok, we compared different Meta-Network architectures (CNN and ViT) and reported the parameter count, FLOPs, and classification accuracy. For the CNN-based Meta-Network, we used our original Meta-Network in the paper since it consists of convolutional layers. The ViT-based Meta-Network consists of six ViTBlocks with layernorm and residual connections. We conducted the experiments on the 16-shot OxfordPet with Stable Diffusion-v2.0.

**Results.** As show in Table 5, both CNN-based and ViT-based Meta-Networks can gain remarkable performance (6-7% accuracy improvement compared with random noise) with only 6-8 M parameters and less than 3 G FLOPs. This indicates that a light Meta-Network with common architectures can accurately learn the noise offset, demonstrating our method is robust and efficient.

## D.3  The Stability of NoOp

**Settings.** To evaluate the stability of NoOp, we randomly sampled three noises (three different seeds) from the Gaussian distribution as the initial noises. Then we conducted the zero-shot classification with them on *CIFAR-10* [36] test set, respectively. Then we used NoOp to optimize them on the 8-shot training set and test. The pretrained diffusion model we used is *Stable Diffusion-v2.0* [35]. We reported the top-1 accuracies before and after training. All other settings are the same as Sec. 4.8.

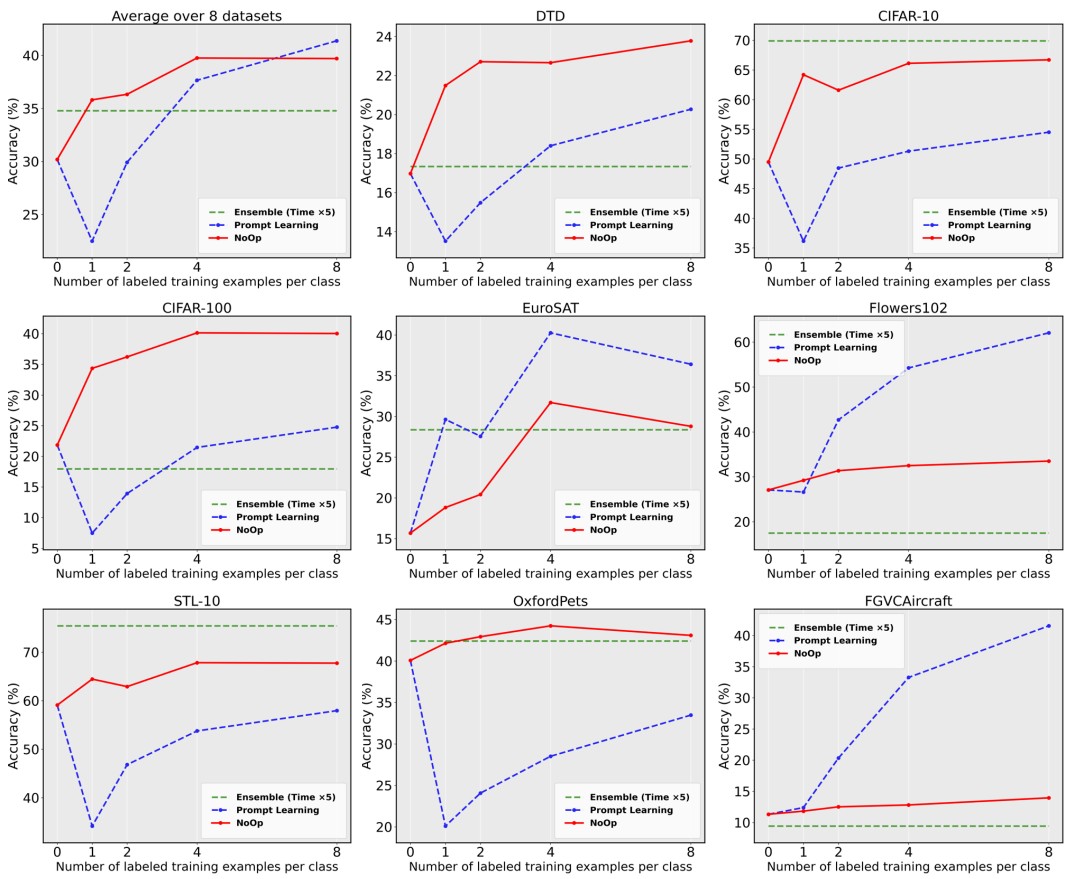

Figure 8: **Additional results of few-shot learning on the eight datasets**. The prompt learning has two problems: (1) It generally harms the performance when the shot number is small (*e.g.*, when the shot number is 1, the prompt learning performs worse than zero-shot DC on six datasets). (2) The variance of prompt learning performance is large (*e.g.*, quite high accuracy improvement for EuroSAT and Flowers102, while quite low for OxfordPets). In contrast, our NoOp can gain more stable and consistent improvements across datasets and shot numbers.

Table 5: The ablation study of the Meta-Network with various architectures.

Figure 9: The stability of NoOp.

| Architecture | Params (M) | FLOPs (G) | Accuracy (%) |
|---|---|---|---|
| random noise | - | - | 40.07 |
| CNN | 7.70 | 2.61 | 47.18 |
| ViT | 6.59 | 2.46 | 46.28 |

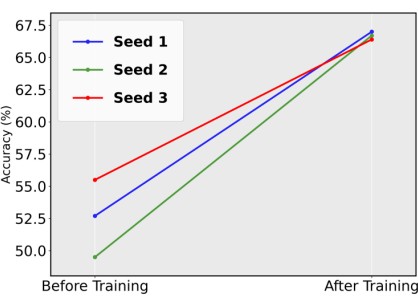

**Results.** As shown in Figure 9, the variance of performances before training is obviously larger than the performances after training. This verified the noise instability problem of DC and the stability of our NoOp, *i.e.*, our method is robust and stable across different random initial noises.

# E    Implementation Details

In this section, we give all the implementation details of our NoOp, the reproduction details, and the computation overhead.

**Details of NoOp**. We used the Discrete Euler as the timestep scheduler, max length padding for the text tokenizer across all the experiments. The initial noise $\epsilon$ is randomly sampled from the standard Gaussian distribution, while the Meta-Network is a U-Net [52] architecture with 3 up-sampling layers and 3 down-sampling layers. Each sampling layer consists of 2-D convolutional layers with ReLu activation [53] and BatchNorm [54]. During training, we used fixed learning rates (w/o warm up and scheduler). The training batch size is 32.

**Other Reproduction Details**. For prompt optimization DC (*c.f.*, Sec. 3.1) implementation in Sec. 4.2 and Sec. D.1, we followed the implementations of TiF Learner [10] and textual inversion [55] respectively.

**Hardware**. All experiments are conducted on 32 NVIDIA V100 GPUs.

**Computation Overhead**. Use 16-shot *CIFAR-10* as an example, the training time of one epoch and the inference time is around 11 seconds and 220 seconds, respectively. This indicates that the noise optimization is quite efficient compared with the high-cost inference.

## F  Limitations and Societal Impacts

**Limitations**. Since our NoOp is still in the framework of the diffusion classifier, there is an inherited limitation: given one image and some category candidates, the inference of NoOp needs to input the image with different categories into the denoising network, respectively. This brings multiple forward propagation (the number of forward propagation equals the number of category candidates) with high computation cost compared to some VLM classifiers (*e.g.*, CLIP only needs to forward one time in its visual encoder for each image).

**Societal Impacts**. On the positive side, as a few-shot learning technique, NoOp can reduce models' reliance on large-scale data, accelerate innovation and technology inclusion, promote applications in various fields, and improve privacy protection and resource efficiency. However, it should be noted that there are potential risks of exacerbating the digital divide and impacting the job market.

