# OpenReview forum: "Noise Matters: Optimizing Matching Noise for Diffusion Classifiers"
_NeurIPS.cc/2025/Conference — NeurIPS 2025 poster_

### Official Review · Reviewer_rqku · 2025-06-01

**Clarity:** 3
**Significance:** 2
**Originality:** 2
**Rating:** 4
**Confidence:** 4

**Summary:**

This paper addresses the instability issue in Diffusion Classifiers, where classification outcomes are sensitive to random noise. The authors propose NoOp, a noise optimization strategy that replaces random sampling with learned “good noise” guided by two principles: frequency matching and spatial matching. A dataset-specific noise and an image-specific noise offset are jointly optimized to stabilize classification. Extensive experiments on multiple datasets show NoOp significantly improves few-shot performance and outperforms costly ensembling baselines. Additionally, NoOp is orthogonal to prompt tuning, allowing further gains. The approach is well-motivated, empirically validated, and shows promise for robust generative classification.

**Questions:**

- The manuscript introduces the principles of “frequency matching” and “spatial matching” as criteria for good noise in diffusion classifiers (DCs), but these are currently heuristically defined and empirically justified. There is no formal theory or precise quantification that explains why these principles are optimal for classification stability or generalization.
- The manuscript compares NoOp mainly with random ensembling and prompt-tuning methods, but omits comparison with other known noise-control or augmentation strategies in generative models (e.g., golden noise, prior-guided sampling, rectified flows with learned priors).
- Typo: line 175 cross-entropy.

**Ethical Concerns:**

["NO or VERY MINOR ethics concerns only"]

**Final Justification:**

Thanks for the authors' rebuttal. The authors have addressed my concerns and I decide to raise my score to 4.

**Limitations:**

yes

**Paper Formatting Concerns:**

No formatting issues.

**Quality:**

3

**Strengths And Weaknesses:**

### Strengths
- The experimental section is thorough, covering multiple datasets, diffusion backbones, and ablation studies. The inclusion of both qualitative and quantitative validations of the proposed principles (frequency and spatial matching) is commendable.
- The writing is clear and easy to follow.

### Weaknesses
- While the idea of optimizing noise is interesting, the method itself is relatively simplistic: it boils down to optimizing a learnable noise vector and a small Meta-Network. This lacks technical novelty or deeper theoretical insight. The two proposed principles (frequency and spatial matching) are intuitive but underdeveloped in formulation and justification.
- The idea of optimizing noise in diffusion models has been explored in generative contexts[1]. The application to classification is somewhat novel, but not fundamentally different in methodology. The contribution feels incremental and may not be strong enough for acceptance at a top-tier venue like NeurIPS.


[1] InitNO: Boosting Text-to-Image Diffusion Models via Initial Noise Optimization, CVPR 2024.

---

> ### Author Rebuttal · Authors · 2025-07-25
>
> Thanks for your thoughtful feedback. We sincerely appreciate your recognition that our experiment is thorough and the paper is well-written.
>
> We hope to address all your concerns below.
>
> ## **Concern 1: "Technical novelty or deeper theoretical insight of simple method"**
>
> **We agree that our method is simple. However, technical novelty is not the contribution that our paper intends to produce, and simplicity $\neq$ no insight**.
>
> Firstly, we clarify that the main contribution of our paper is that we analyze the role of noise in the current diffusion classifier (DC). That is, the noise destroys some parts of the image, and DC tries to find the category that can best guide the diffusion model to reconstruct the destroyed parts. And this analysis of the noise role can exactly explain why there is a Noise Instability problem in DC. Then, based on the role analysis, we proposed the two principles (Frequency Matching and Spatial Matching) to guide the noise to destroy the image. **The above analysis of the noise role, the noise instability problem, and guidance for noise optimization constitutes the insights of our paper and derives the design of our method.**
>
> Secondly, although the framework of our method is simple, the results show its effectiveness with consistent improvements. **Simple but effective**, it exactly indicates that we catch the key research gap that exists in the Diffusion Classifier.
>
>
> Thirdly, there are a lot of **advantages of the simple method**. For example, simple and efficient algorithms are easier to implement in large-scale industries and for subsequent research expansion.
>
>
> Similarly, [1, 2, 3, 4, 5] are some related and representative works with simple and effective methods.
>
>
> ---
>
> ## **Concern 2: Formulation and justification for the two proposed principles (Frequency and Spatial Matching)**
>
> Thanks for the reviewer's insightful feedback.
>
> Firstly, by analyzing the role of noise in the Diffusion Classifier, i.e., destroying the category-related signals, it is natural to target the specific range of frequency space and spatial space. For the frequency space, there are some works [6, 7] that have discussed the category-related signals are located in the dataset-specific range. **The Frequency Bias Theory [6, 7] can be a justification of our proposed Frequency-Matching principle (we will supplement it in the future version).** For the Spatial Matching, it's intuitive that even for images within one dataset, the category-related signals should be located in different areas. Thus, we proposed another Spatial Matching principle. These two principles guide us to design the optimization framework with a dataset-specific learnable noise and a Meta-Network to produce the image-specific noise bias.
>
> Secondly, as the reviewer mentioned that "both qualitative and quantitative validations of the proposed principles (frequency and spatial matching) is commendable." These results can verify that **the optimization can actually align with these two principles** and provide some empirical justifications.
>
> Finally, we agree with the reviewer that a stricter formulation or justifications are required to prove that these two principles are optimal for classification stability and generalization. However, we do not overclaim that our proposed principles are optimal because the optimal principles for classification are a general and open question in this field. Instead, we claim that **they are two typical and intuitive principles that can guide an effective optimization framework design**.
>
> ---
>
> ## **Concern 3: Difference and contribution of our method compared with the current optimizing noise in the generation field**
>
> We apologize for any confusion.
>
> As we discussed in Lines 99-107 of our paper, the noise optimization and selection have been widely explored in the generation scope [4, 5, 8, 9, 10, 11, 12, 13]. However, in the classification domain, the motivation and optimization guidelines are significantly different from the generation domain. We clarify the differences and contributions of our work below.
>
> ### **Different motivations** ###
>
> To the best of our knowledge, we are the first paper to focus on the noise optimization for the discriminative task. The noise instability is a very severe problem that decreases the inference stability and efficiency in Diffusion Classifier and even more general diffusion-based perception scenarios. Our motivation is to reveal the research gap, and one intuitive and effective method to mitigate this problem is noise optimization.
>
> ### **Different optimization guidelines** ###
>
> For the generation domain, the guideline or the final goal of noise optimization is to control the generation process and get the desired synthetic images. Based on this guideline, the statistics-based, attention-based, and VLM-based criteria were proposed to guide the optimization and find a good starting point for the desired generation. In contrast, the guideline for our method is derived from the role of noise played in the Diffusion Classifier, i.e., destroying category-related signals of the image. We argue that a good noise for the Diffusion Classifier can destroy the target part of the images and then fully unleash the diffusion model's reconstruction difference under different category conditions. Based on this guideline, we propose two principles (Frequency Matching and Spatial Matching) to guide the corresponding noise optimization framework design.
>
> ### **Technical gaps** ###
>
> There are technical gaps for noise optimization between generation and classification. For example, leveraging Z-normalization to scale the logits difference and maintain the gradient during training (c.r.f., Lines 166-173).
>
> ### **Our contribution** ###
>
> Our contribution is analysing the role of noise in the Diffusion Classifier and revealing the Noise Instability problem. To mitigate this problem, we proposed two principles to guide the noise optimization and derive the design an effective method.
>
> ---
>
> ## **Concern 4: Comparison with methods like noise-control or augmentation strategies in generative models**
>
> The noise-control or augmentation strategies like golden noise [5] and rectified flows with learned priors [14] mainly focus on the optimization objective for generation. However, there is an **objective gap between generation and classification**.
>
> For the generation task, the objective can be maximizing the preference score or minimizing the RFDS (Rectified Flow Distillation Sampling) loss [14]. However, the optimization objective of classification is to maximize the reconstruction difference under different category conditions.
>
> With different optimization objectives, they can not be comparable. Moreover, apart from the optimization objectives, the noise optimization frameworks of [5, 14] are too simple (directly optimizing an NPNet or parameterized noise) to be compared as baselines because their main significant contributions are to find effective optimization objectives instead of optimization method.
>
>
>
> ---
>
> ### References:
>
>
> [1] Zhou, Kaiyang, et al. "Learning to prompt for vision-language models." International Journal of Computer Vision 130.9 (2022): 2337-2348.
>
> [2] Zhou, Kaiyang, et al. "Conditional prompt learning for vision-language models." Proceedings of the IEEE/CVF conference on computer vision and pattern recognition. 2022.
>
> [3] Ma, Nanye, et al. "Inference-time scaling for diffusion models beyond scaling denoising steps." arXiv preprint arXiv:2501.09732 (2025).
>
> [4] Li, Shuangqi, et al. "Enhancing compositional text-to-image generation with reliable random seeds." The Thirteenth International Conference on Learning Representations. 2024.
>
> [5] Zhou, Zikai, et al. "Golden noise for diffusion models: A learning framework." arXiv preprint arXiv:2411.09502 (2024).
>
> [6] Lin, Zhiyu, Yifei Gao, and Jitao Sang. "Investigating and explaining the frequency bias in image classification." arXiv preprint arXiv:2205.03154 (2022).
>
> [7] Wang, Shunxin, et al. "What do neural networks learn in image classification? a frequency shortcut perspective." Proceedings of the IEEE/CVF International Conference on Computer Vision. 2023.
>
> [8] Guo, Xiefan, et al. "Initno: Boosting text-to-image diffusion models via initial noise optimization." Proceedings of the IEEE/CVF Conference on Computer Vision and Pattern Recognition. 2024.
>
> [9] Wang, Ruoyu, et al. "The silent prompt: Initial noise as implicit guidance for goal-driven image generation." arXiv e-prints (2024): arXiv-2412.
>
> [10] Chen, Changgu, et al. "Find: Fine-tuning initial noise distribution with policy optimization for diffusion models." Proceedings of the 32nd ACM International Conference on Multimedia. 2024.
>
> [11] Miao, Boming, et al. "Noise Diffusion for Enhancing Semantic Faithfulness in Text-to-Image Synthesis." Proceedings of the Computer Vision and Pattern Recognition Conference. 2025.
>
> [12] Qi, Zipeng, et al. "Not all noises are created equally: Diffusion noise selection and optimization." arXiv preprint arXiv:2407.14041 (2024).
>
> [13] Samuel, Dvir, et al. "Generating images of rare concepts using pre-trained diffusion models." Proceedings of the AAAI Conference on Artificial Intelligence. Vol. 38. No. 5. 2024.
>
> [14] Yang, Xiaofeng, et al. "Text-to-image rectified flow as plug-and-play priors." arXiv preprint arXiv:2406.03293 (2024).

---

### Official Review · Reviewer_a2bg · 2025-07-02

**Clarity:** 3
**Significance:** 2
**Originality:** 2
**Rating:** 3
**Confidence:** 3

**Summary:**

This paper tackles the noise instability problem in diffusion classifiers (DCs), where the randomness of Gaussian noise injected during the forward process leads to high variance in classification outcomes. To address this, the authors propose a noise optimization method (NoOp) that learns to selectively destroy category-relevant frequency components (frequency matching) and spatial regions (spatial matching). They introduce a learnable dataset-specific noise and a meta-network that predicts image-specific noise offsets. The sum of these two replaces the standard random noise in DCs. Through extensive experiments, the authors demonstrate that this optimized noise both stabilizes and improves classification performance, eliminating the need for costly multi-sample ensembling.

**Questions:**

Please see weaknesses

**Ethical Concerns:**

["NO or VERY MINOR ethics concerns only"]

**Final Justification:**

The authors have attempted to address my concerns, but the theoretical justification remains insufficient, and the performance gains on VBench metrics are not adequately substantiated. Therefore, I will maintain my score as borderline reject.

**Limitations:**

Yes

**Quality:**

2

**Strengths And Weaknesses:**

**Strengths**
- The paper is well written and easy to follow.
- The authors provide comprehensive experiments on eight datasets and SD/ReFlow models. They also provide various ablation studies and compatibility with the prompt optimization method.

**Weaknesses**
- By using NoOp to produce a deterministic, optimized noise at a fixed timestep (e.g., $t=500$), the method intuitively resembles the effect of ensembling over multiple random noises. It would be better if the authors could provide a theoretical analysis to guarantee that this deterministic approach approximates the expectation of classification under noise sampling.

- The reported performance in this paper appears significantly different from baselines [1], which raises concerns about the consistency and comparability of the experiments. Moreover, because NoOp is trained explicitly on a particular dataset and at a single timestep, its generalization ability is questionable, especially when compared to vision-language model (VLM) based classifiers that benefit from richer semantic priors and typically achieve stronger cross-dataset performance. By relying on a fixed, optimized noise, this approach also sacrifices the inherent generative and stochastic diversity of diffusion models, which further limits its broader appeal.

[1] Clark et al., Text-to-Image Diffusion Models are Zero-Shot Classifiers, NeurIPS'23.

---

> ### Author Rebuttal · Authors · 2025-07-25
>
> We thank the reviewer for the constructive feedback! We hope to address concerns below.
>
>
> ## **Concern 1: "Theoretical analysis to guarantee that this deterministic approach approximates the expectation of classification under noise sampling"**
>
> We apologize for any confusion.
>
> Here we clarify that the Ensembling (expectation of classification under noise sampling) and our Noise Optimization improve the classification performance with **different mechanisms**.
>
> ### **Why does Ensembling work?**
>
> Let the $m$ base predictors be
>
> $\hat{h}_i(x)=f(x)+\varepsilon_i(x), \qquad i=1,\dots,m$,
>
> where $f(x)$ is the true function and the noises $\varepsilon_i(x)$ are i.i.d. such that
>
> $\mathbb{E}[\varepsilon_i(x)] = 0, \qquad \operatorname{Var}[\varepsilon_i(x)] = \sigma^{2}.$
>
> We define their expectation (simple average)
>
>
> $\bar{h}(x)=\frac{1}{m}\sum_{i=1}^{m}\hat{h}_i(x).$
>
> Then we compare the mean squared error (MSE) of a single model with the ensemble model.
>
> For bias,
>
> $\mathbb{E}[\hat{h}_i(x)] = f(x)
> \quad\text{and}\quad
> \mathbb{E}[\bar{h}(x)] = f(x).$
>
> Both the single model and the ensemble are unbiased.
>
> For variance,
>
> $
> \operatorname{Var}\\bigl[\bar{h}(x)\bigr]
> = \operatorname{Var}\\left(
>       \frac{1}{m}\sum_{i=1}^{m}\varepsilon_i(x)
>   \right)
> = \frac{1}{m^{2}}\sum_{i=1}^{m}\operatorname{Var}[\varepsilon_i(x)]
> = \frac{\sigma^{2}}{m}.
> $
>
> With squared‑error loss, $\operatorname{MSE}=\text{Bias}^2+\text{Variance}$.
> Because the biases are identical, the ensemble’s smaller variance yields
>
>
> $
> \operatorname{MSE}\bigl[\bar h(x)\bigr]
>   = \frac{\sigma^{2}}{m}
>   <  \sigma^{2}
>   = \operatorname{MSE}\bigl[\hat h_i(x)\bigr], m \ge 2.
> $
>
> Hence, averaging on multiple sampled noises strictly decreases expected risk.
>
> ### **Why does Noise Optimization work?**
>
> To answer this question, we recall the mechanism of the Diffusion Classifier (DC).
>
> Given an inference image, the DC firstly adds some noise to destroy some of the signals that are related to the category. Then, using different categories as the prompts to reconstruct the noisy image. The final prediction is the category that can reconstruct it the best.
>
> In this pipeline, a very crucial part is adding noise because it decides which signals are removed by adding noise. The best situation is trying to remove the category-related signals and maintain other category-independent signals. In that case, the reconstruction difference of diffusion models under different prompts can be fully unleashed.
>
> Thus, our method can **learn an optimized noise to target the category-related signals**. This kind of optimized noise is usually in the low-probability areas of the standard Gaussian distribution, and thus is very difficult to sample. In comparison, the randomly sampled noises are usually in the high-probability areas of the standard Gaussian distribution but lack the target-destroying capacity.
>
>
> ### **Additional evaluation to justify why Noise Optimization works**
>
> Here we compared the random noise and the optimized noise and reported their mean, variance, and standard Gaussian distribution probability density (pdf).
>
> |       | mean | varience  | pdf (log) |
> | :---        |    :----:   |          :---: |         :---: |
> | random noise      | -0.0032       | 1.0021   | -23265 |
> | optimized noise  | -0.0043       | 1.0282      | -23479 |
>
>
> We can see that, compared with the random sampled noise, the optimized noise's mean is farther away from $0$, variance is farther away from $1$, and the probability density is lower. This demonstrated that the **optimized noise is usually in the low-probability areas of the standard Gaussian distribution to target the category-related signals**. And this kind of noise is difficult to sample. Thus, we obtain it by optimization.
>
> **Thus, the optimized noise does not try to approximate the expectation of multiple randomly sampled noises. Instead, it improves the classification performance through a different mechanism. That is, it targets and destroys the category-related signals. **
>
> ---
>
> ## **Concern 2: Performance difference from baseline [1]**
>
> We apologize for any confusion.
>
> Here, we explain why our reported performance is different from baseline [1].
>
> We take CIFAR-10 with Stable Diffusion-v1.4 as an example. In our paper, we reported that the accuracy is 67.59% (c.r.f.,  the blue dashed line in Figure 4, CIFAR-10). However, in Table 1 of [1], the CIFAR-10 accuracy is 72.1%.
>
> **The reason for this inconsistency is that we only ensemble with five different noises while [1] uses more ensemble noises (such as 20+).** The specific number of ensemble scale for different datasets is dynamically decided by its pruning operation (c.r.f., Algorithm 1 in [1]). More noises are used for ensemble, resulting in higher inference cost and better performance.
>
>
> For our paper, we focus on noise optimization. Thus, for the expensive ensembling method, we only use five noises for ensembling.
>
> ---
>
> ## **Concern 3: Evaluation for cross-dataset generalization**
>
> We thank the reviewer for valuable questions.
>
> In response to the question of whether our method can achieve similar cross-dataset generalization to vision-language models (VLMs), we evaluated our method for cross-dataset generalization settings.
>
> Specifically, we optimized the noise and the Meta-Network on 4-shot ImageNet (source dataset) and tested it on the source dataset and eight target datasets. We compared our method with the ensembling methods (5 times computation), and Δ denotes our method's gain over ensembling.
>
> |             |  &nbsp;&nbsp;Source |  &nbsp;&nbsp;&nbsp;&nbsp;&nbsp;&nbsp;   |      |           |             |             |   &nbsp;&nbsp;&nbsp;Target       |       |         |         |         |         |         |
> | :------- |   :----:   | :----: | :----: | :----: | :----: | :----: | :----: | :----: | :----: | :----: | :----: | :----: | :----: |
> |            | ImageNet | |DTD | CIFAR10 | CIFAR100 | EuroSAT | Flowers102 | STL10 | OxfordPets | Aircraft | Average |
> | Ensemble (Time ×5) | 25.94 | | 17.34 | **69.91** | 17.96 | 28.37 | 17.47 | **75.44** | 42.41   | 9.42 | 34.79 |
> | Ours     | 26.34   | |**21.70**  | 63.26 | **29.36**   | **29.31**  | **29.48**    | 71.66 | **45.90** | **10.56**  | **37.65** |
> | Δ          | *+0.40* | | *+4.36*  | *-6.65*   |  *+11.4*    |  *+0.94*      | *+12.01* | *-3.78* | *+3.49* | *+1.14* | *+2.86* |
>
> As we can see, the noise and Meta-Network learned on ImageNet can also be used in the eight target datasets. Specifically, six of eight can outperform the ensembling methods, and our method can improve 2.86% accuracy on average.
>
> This demonstrated that the noise optimization can learn some generalized knowledge that is beneficial to the classification, i.e., how to destroy the target part of the image and better utilize the reconstruction capacity to achieve classification. **To this end, this result indicates that although NoOp was trained on a particular dataset and at a single timestep, it can also be generalized to other datasets, showing remarkable generalization ability.**
>
> ---
>
> ## **Concern 4: Sacrificing the stochastic diversity of diffusion models**
>
> We agree with the reviewer's perspective, that is, the fixed and optimized noise will sacrifice the stochastic diversity of diffusion models.
>
> However, we clarify that diversity is not always a positive property across tasks.
>
> ### **Diversity in generation task**
> For the generation task, sampling different noises as the initial point can lead to diverse generation results. This satisfied the requirements of humans regarding to generation application.
>
> ### **Diversity in classification task**
> For the classification task, we use the Diffusion Classifier as an example. The stochastic diversity of different noise will lead to the Noise Instability problem (c.r.f., Figure 1 (b) in our paper). In that case, the stochastic diversity is negative for classifying an image.
>
> ### **Ensembling the optimized noises across timesteps**
>
> Although our method optimizes to a fixed noise for a given timestep, it can still leverage the diverse knowledge across timesteps. To be specific, for different timesteps, we can optimize different noises. This enables our method to leverage knowledge of the diffusion model conditioned on different timesteps.
>
> We also conducted experiments for ensembling our NoOp across different timesteps in the Section. A.3.3 of the appendix. The results indicated that NoOp can optimize different noises for different timesteps and ensemble the prediction to gain a further performance improvement.
>
>
> ---
>
> ### References:
>
> [1] Clark, Kevin, and Priyank Jaini. "Text-to-image diffusion models are zero shot classifiers." Advances in Neural Information Processing Systems 36 (2023): 58921-58937.

---

> > ### Author Response · Authors · 2025-08-08
> >
> > Dear Reviewer a2bg,
> >
> > We sincerely appreciate the time and effort you have dedicated to reviewing our paper.
> >
> > As the discussion period approaches its end, we would like to gently remind you that there is one day remaining for additional questions. We would be grateful for the opportunity to address any further concerns you may have before the discussion phase concludes.
> >
> > Thank you very much!
> >
> > The Authors

---

### Official Review · Reviewer_CARn · 2025-07-03

**Clarity:** 3
**Significance:** 2
**Originality:** 3
**Rating:** 4
**Confidence:** 4

**Summary:**

In order to circumvent the tradeoff between the unstable single-sample sampling and expensive multiple-sample ensembling for current Diffusion Classifier (DC) frameworks, this paper proposes a new noise optimization method named NoOp to predict the 'good noises' for classification. Such framework consists of two parts: (1) a dataset-specific learnable noise for frequency matching; and (2) a Meta-Network designed for predicting the image-specific noise offset for spatial matching. Given a pre-trained diffusion model and a K-way-N-shot image classification task, the learnable noise and the Meta-Network is jointly optimized using the cross-entropy loss with Z-score normalization on the training samples and then evaluated on the full test set.
Extensive experiments conducted on 3 different diffusion models and 8 different classification datasets validate the effectiveness of this method.

**Questions:**

Please refer to the weaknesses.

**Ethical Concerns:**

["NO or VERY MINOR ethics concerns only"]

**Final Justification:**

Since most of my concerns are addressed by the authors, I decided to keep my score as 4 and am inclined to accept this paper.

**Limitations:**

yes

**Quality:**

2

**Strengths And Weaknesses:**

Strengths:
1) The method is simple.
2) The experiments are extensive.
3) The presentation is coherent and the writing is easy to follow.

Weaknesses:


1) The detailed architecture design of the Meta-Network seems missing. Why using a U-Net architecture rather than using a simple CNN or ViT for such a 'light' Network? What is the parameter count and FLOPs?

 2) The motivation behind the frequency matching noise for each dataset is a little confusing. From my perspective, it learns the joint noise bias for the whole dataset, which means it can be directly absorbed into the Meta Network as a final learnable element-wise bias parameter. Why separate it out and make the whole pipeline more complicated?

3) I am also wondering how to apply such a framework to open-set classification problem. Or in other words, how to design the frequency matching term if I want to train one single classifier for all of the datasets included in this paper?

---

> ### Author Rebuttal · Authors · 2025-07-25
>
> We sincerely appreciate your constructive feedback, and thanks for finding that our method is simple and the paper is well-written.
>
> We hope to address all your concerns below.
>
> ---
>
>
>
> ## **Concern 1: The detailed architecture design of the Meta-Network and the ablation study of the Meta-Network with various architectures**
>
> We apologize for any confusion.
>
> ### **The details of Meta-Network**
>
> The Meta-Network is a U-Net architecture with 3 up-sampling layers and 3 down-sampling layers. Each sampling layer consists of 2-D convolutional layers with ReLu activation and BatchNorm (c.f., more details can be found in Lines 108-110 of the appendix and the source code). As the basic module of the U-Net is the convolutional layer, our Meta-Network can be considered as a simple CNN network.
>
> The reason why we use a U-Net architecture is that the Meta-Network's input (image) and output (noise offset) share the same shape. For this kind of scenario, U-Net is a common architecture to maintain the shape.
>
> ### **Additional Meta-Network Ablation**
>
>
> To respond to your question about the ablation of various architectures, parameter count, and FLOPs. We compared different Meta-Network architectures (CNN and ViT) and reported the parameter count, FLOPs, and classification accuracy. For the CNN-based Meta-Network, we used our original Meta-Network in the paper since it consists of convolutional layers. The ViT-based Meta-Network consists of six ViTBlocks with layernorm and residual connection. We conducted the experiments on the 16-shot OxfordPet with Stable Diffusion-v2.0. Other settings are the same as Section. 4.1.
>
>
> | Architecture      | Params (M) | FLOPs (G)| Accuracy (%) |
> | :---        |    :----:   |          :---: |          :---: |
> | random noise    | - | - | 40.07 |
> | CNN    | 7.70 | 2.61 | 47.18 |
> | ViT   | 6.59 | 2.46 | 46.28 |
>
> As we can see, both CNN-based and ViT-based Meta-Networks can gain remarkable performance (6-7% accuracy improvement compared with random noise) with only 6-8 M parameters and less than 3 G FLOPs. This indicates that a light Meta-Network with common architectures can accurately learn the noise offset, demonstrating our method is robust and efficient.
>
> ---
>
> ## **Concern 2: Motivation behind the Frequency Matching noise for each dataset**
>
> We apologize for any confusion and clarify our motivation for Frequency Matching noise below.
>
> ### **Leveraging the prior that images within one dataset share a common frequency bias.**
>
> We agree with the reviewer's perspective: "Noise bias can be directly absorbed into the Meta Network as a final learnable element-wise bias parameter." However, we already have a prior that category-related signals of images within one dataset usually share a frequency range. Based on this prior, we can derive that there is a common noise bias for all images within this dataset. Then, we parametrized this common noise bias as the learnable noise to reduce the learning difficulty of the Meta-Network. In that case, the training can be more stable and converge to a better point compared with only training the Meta Network to learn element-wise bias.
>
> ### **A high-level perspective**
>
> In theory, the Meta-Network can satisfy both Frequency Matching and Spatial Matching. But it's more difficult to learn the dataset-specific bias and image-specific bias at the same time. If we add another component to learn the dataset-specific bias, the task will be easier for the Meta-Network (i.e., only focus on learning image-specific bias), making the training more stable.
>
> ### **The experimental perspective**
>
> We conducted the ablation study in Figure 6. We can see that jointly training a learnable noise (Frequency Matching) and a Meta-Network (Spatial Matching) is more stable and can achieve a better performance compared with only training a Meta-Network. This result can support the significance of optimizing a shared noise for the Frequency Matching.
>
> ---
>
> ## **Concern 3: Extending to open-set scenarios**
>
> Thanks for your insightful question.
>
> To respond to the reviewer's question about how to apply our method to open-set classification, we tested our method in the cross-dataset transfer setting. To be specific, we optimized the noise and the Meta-Network on 4-shot ImageNet (source dataset) and tested on eight target datasets. This can be considered as an open-set problem since a lot of categories in the target datasets did not appear in the source dataset. We compared our method with the ensembling methods (5 times computation), and Δ denotes our method's gain over ensembling.
>
> |             |  &nbsp;&nbsp;Source |  &nbsp;&nbsp;&nbsp;&nbsp;&nbsp;&nbsp;   |      |           |             |             |   &nbsp;&nbsp;&nbsp;Target       |       |         |         |         |         |         |
> | :------- |   :----:   | :----: | :----: | :----: | :----: | :----: | :----: | :----: | :----: | :----: | :----: | :----: | :----: |
> |            | ImageNet | |DTD | CIFAR10 | CIFAR100 | EuroSAT | Flowers102 | STL10 | OxfordPets | Aircraft | Average |
> | Ensemble (Time ×5) | 25.94 | | 17.34 | **69.91** | 17.96 | 28.37 | 17.47 | **75.44** | 42.41   | 9.42 | 34.79 |
> | Ours     | 26.34   | |**21.70**  | 63.26 | **29.36**   | **29.31**  | **29.48**    | 71.66 | **45.90** | **10.56**  | **37.65** |
> | Δ          | *+0.40* | | *+4.36*  | *-6.65*   |  *+11.4*    |  *+0.94*      | *+12.01* | *-3.78* | *+3.49* | *+1.14* | *+2.86* |
>
>
> From the results, the noise and Meta-Network learned on ImageNet can also be used in the eight target datasets. Specifically, six of eight can outperform the ensembling methods, and our method can improve 2.86% accuracy on average.
>
> **This demonstrated that the noise optimization can learn some generalized bias that is beneficial to the classification on one general source dataset and then generalize to different target datasets. To this end, this result indicates that our method is robust and can be applied in open-set generalization scenarios.**

---

### Official Review · Reviewer_Drnp · 2025-07-06

**Clarity:** 3
**Significance:** 3
**Originality:** 3
**Rating:** 5
**Confidence:** 4

**Summary:**

This paper introduces a method that enhances diffusion-based open-set classification by learning image- or class-specific noise offsets from a support set via a Meta-Network. The approach modifies the original diffusion classifier framework by incorporating a training phase based on support examples. The method is evaluated on few-shot classification tasks, with comparisons to baseline models. Several aspects such as training cost, scalability with respect to the number of classes, and reliance on support data are identified as important factors for further analysis.

**Questions:**

1. The method involves learning noise offsets from a support set and training a Meta-Network. In scenarios with a large number of classes, is this approach still scalable? Specifically, can the method effectively learn discriminative noise signals when the class space is large, and does the training cost grow significantly with the number of classes? A discussion or experiment on scalability would be informative.
2. Since the method learns noise from a small support set, is there a risk of distribution shift between the support and test data during inference, especially in open-world settings? It would be helpful if the authors could evaluate the method on open-set recognition or under domain shifts to better understand its generalization.
3. The original Diffusion Classifier suffers from high computational cost and slow inference speed. Given that the proposed method introduces additional training (e.g., Meta-Network optimization), how does the overall training and inference cost compare with the original DC and other few-shot or zero-shot methods? Does the method improve or worsen the efficiency bottleneck of DC?

**Ethical Concerns:**

["NO or VERY MINOR ethics concerns only"]

**Final Justification:**

My concerns are well addressed and I will keep the score.

**Limitations:**

Yes

**Quality:**

3

**Strengths And Weaknesses:**

Strengths:
1. The idea of optimizing noise for diffusion classifiers is novel and intuitive.
2. The method is simple and efficient, requiring only a lightweight Meta-Network and a noise offset, and compatible with existing diffusion backbones.
3. The experiments are comprehensive, covering multiple datasets and model variants.

Weaknesses:
1. Diffusion classifiers were originally designed for zero-shot classification, but the proposed method relies on few-shot training to optimize the noise. This compromises the zero-shot capability of DC and may limit its applicability in fully open-world or zero-data scenarios.
2. Although the experiments are extensive, a cross-dataset evaluation would better demonstrate the generalization ability of the proposed method.
3. There may be some format problems. Captions show below the figure, etc.

---

> ### Author Rebuttal · Authors · 2025-07-25
>
> Thanks for your valuable feedback on our work. We sincerely appreciate your recognition of our method as both novel and effective, as well as the comprehensive evaluations.
>
> We hope to address all your concerns below.
>
> ---
>
> ## **Concern 1: Evaluation for open-world generalization**
>
> Thank you for your valuable suggestions.
>
> In response to your feedback, we evaluated our method in the cross-dataset transfer setting to demonstrate its generalization ability on open-set recognition. We optimized the noise and the Meta-Network on 4-shot ImageNet (source dataset) and tested it on the source dataset and eight target datasets. We compared our method with the ensembling methods (5 times computation), and Δ denotes our method's gain over ensembling.
>
> |             |  &nbsp;&nbsp;Source |  &nbsp;&nbsp;&nbsp;&nbsp;&nbsp;&nbsp;   |      |           |             |             |   &nbsp;&nbsp;&nbsp;Target       |       |         |         |         |         |         |
> | :------- |   :----:   | :----: | :----: | :----: | :----: | :----: | :----: | :----: | :----: | :----: | :----: | :----: | :----: |
> |            | ImageNet | |DTD | CIFAR10 | CIFAR100 | EuroSAT | Flowers102 | STL10 | OxfordPets | Aircraft | Average |
> | Ensemble (Time ×5) | 25.94 | | 17.34 | **69.91** | 17.96 | 28.37 | 17.47 | **75.44** | 42.41   | 9.42 | 34.79 |
> | Ours     | 26.34   | |**21.70**  | 63.26 | **29.36**   | **29.31**  | **29.48**    | 71.66 | **45.90** | **10.56**  | **37.65** |
> | Δ          | *+0.40* | | *+4.36*  | *-6.65*   |  *+11.4*    |  *+0.94*      | *+12.01* | *-3.78* | *+3.49* | *+1.14* | *+2.86* |
>
> As we can see, the noise and Meta-Network learned on ImageNet can also be used in most of the target datasets. Specifically, six of eight can outperform the ensembling methods, and our method can improve 2.86% accuracy on average.
>
> **This demonstrated that the noise optimization can learn some generalized knowledge that is beneficial to the classification, i.e., how to destroy the target part of the image and better utilize the reconstruction capacity to achieve classification. To this end, this result indicates that our method can be further leveraged in open-world scenarios.**
>
> ---
>
> ## **Concern 2: Scalability to scenarios with a large number of classes**
>
> Thank you for your insightful question.
>
> In the original paper, we conducted experiments on eight datasets. However, the dataset with the largest class number is Flowers102 (102 classes). The reviewer suggested we verify whether our method can be scaled to scenarios with a large number of classes. Thus, we experimented on ImageNet (1000 classes, nearly 10 times more than Flowers102). We used the same setting as Section. 4.1 with Stable Diffusion-v2.0 and only replace the dataset with ImageNet.
>
>
> |       | Ensemble (Time ×5) | shot=0 | shot=1 | shot=2 | shot=4 | shot=8| shot=16 |
> | :---        |    :----:   |          :---: |         :---: |         :---: |         :---: |         :---: |         :---: |
> | Accuracy (%)   |   25.94    |   17.34    | 24.62 | 25.31 | 26.34 | 27.59 | 29.13 |
>
> **This result indicated that our method can be applied to scenarios with a large number of classes, even with only an optimized noise and a small Meta-Network.** Our method can outperform the ensemble baseline with only four shots, and the accuracy improves as the number of shots increases.
>
> ---
>
> ## **Concern 3: Computational cost analysis and comparisons**
>
> As the reviewer mentioned, the original Diffusion Classifier suffers from high computational cost and slow inference speed, and whether our method can improve or worsen the efficiency. It is worth mentioning that **our method can significantly improve the overall efficiency**. For this concern, we split into two parts (formal cost analysis and experimental comparisons) to respond.
>
> ### **Formal cost analysis**
> Given a dataset $\mathcal{D}$ with $K$ categories. We assume that the number of training samples for each category (i.e., the shot number) is $N$. The size of the test set is $M$ (i.e., $M$ test samples).
>
> In that case,
>
> Training Time $ \propto KN$,
>
> Test Time $ \propto KM$.
>
> Usually, $N=1,2,4,8,16$ for few-shot learning is significantly smaller than $M$. Thus, the training time of our method can be ignored compared with the test time. However, after this low-cost training, the sampling times of noises can be significantly decreased to improve the efficiency of the original Diffusion Classifier.
>
> ### **Experimental comparisons**
>
> We used ImageNet as an example to demonstrate how our method can improve the efficiency of the original Diffusion Classifier. The unit of time cost is one hour with 32 NVIDIA V100 GPUs. For the ensemble method, we randomly sample five different noises.
>
> | Method      | Training Time | Inference Time | Total Time | Accuracy (%) |
> | :---        |    :----:   |          :---: |          :---: |          :---: |
> | Ensemble (Time ×5)  | 0 | 153.0 | 153.0 | 25.94 |
> | Shot = 0   | 0 | 30.6 | 30.6 | 17.34 |
> | Shot = 1   | 0.4 | 30.6 | 31.0 | 24.62 |
> | Shot = 2   | 0.8 | 30.6 | 31.4 | 25.31 |
> | Shot = 4   | 1.6 | 30.6 | 32.2 | 26.34 |
> | Shot = 8   | 3.2 | 30.6 | 33.8 | 27.59 |
> | Shot = 16 | 6.4 | 30.6 | 37.0 | 29.13 |
>
> As we can see, our training time is quite smaller than the inference time. Even for the 16-shot training, the total time is only around 24% while the accuracy gains 3.19% compared with the 5-times ensembling method.

---

> > ### Comment · Reviewer_Drnp · 2025-08-05
> >
> > Thank you for the reply. My concerns are well addressed.

---

### Note · Authors · 2025-08-13

We sincerely appreciate all reviewers for their constructive feedback. We are encouraged that they found our work is **well-motivated** [rqku], and our idea of optimizing noise for diffusion classifiers is **novel and intuitive** [Drnp]. Meanwhile, we are glad they think the paper is **coherent** [CARn], **well written** [rqku], and **easy to follow** [CARn, a2bg, rqku]. Additionally, our **extensive** [Drnp, CARn, a2bg, rqku] experiments and **commendable** [rqku] qualitative and quantitative validations of the proposed principles verify that our method is **simple yet effective** [Drnp, CARn]  and of **compatibility** [Drnp, a2bg].

We have revised our paper according to the comments. The major changes are summarized as follows:

- According to Reviewer **Drnp**’s comments:
    - Experiment. We add experiments on open-world generalization and datasets with a large number of classes.
    - Analysis. We formally analyze the computational cost and report the specific time cost comparisons.

- According to Reviewer **CARn**‘s comments:
    - Experiment. We add an ablation study on the architectures of Meta-Network and experiments on open-set classification.
    - Clarification. We offer a clarification of the motivation behind the Frequency Matching noise.

- According to Reviewer **a2bg**’s comments:
    - Analysis. We analyze the different mechanisms between ensembling and our noise optimization with both theoretical and empirical justification.
    - Clarification. We offer clarifications of the performance difference and the necessity of stochastic diversity in classification tasks.
    - Experiment. We add experiments on cross-dataset generalization.

- According to Reviewer **rqku**’s comments:
    - Clarification. We offer clarifications of our main contribution and the difference from noise optimization in generation.
    - Justification. We provide the justification of the two proposed principles with related works and theoretical basis.
    - Comparison. We compare the noise optimization objective between generation and classification and demonstrate that the optimization methods across tasks can not be comparable.



After providing the above revisions, three reviewers [Drnp, CARn, rqku] acknowledged the rebuttal, and two of them [Drnp, rqku] commented that **concerns are addressed.** Meanwhile, one reviewer [a2bg] did not respond to us. Thanks again for all the feedback.


Many thanks,

The Authors

---

### Decision · Program_Chairs · 2025-09-17

**Decision:**

Accept (poster)

**Comment:**

This paper presents a simple yet effective method to optimize the noise used to destroy the image in diffusion classifier, which tends to  be unstable and requires ensembling. The reviewers find this paper well-motivated [rqku], intuitive [Drnp], and easy to follow [CARn, a2bg, rqku]. They also appreciate the extensive [Drnp, CARn, a2bg, rqku] experiments to validate the proposed insights empirically. Reviewer a2bg expressed concerns on the theoretical justification and the complexity of the method. After considering all the reviews and rebuttal, AC decides that the merits / insights of this paper outweigh the concerns of reviewer a2bg and recommends acceptance.